# Oolong: Investigating What Makes Transfer Learning Hard with Controlled Studies

**Zhengxuan Wu**[*], **Alex Tamkin**[*], **Isabel Papadimitriou**[*†]

Stanford University

{wuzhengx, atamkin, isabelvp}@stanford.edu

## Abstract

When we transfer a pretrained language model to a new language, there are many axes of variation that change at once. To disentangle the impact of different factors like syntactic similarity and vocabulary similarity, we propose a set of *controlled transfer studies*: we systematically transform the language of the GLUE benchmark, altering one axis of crosslingual variation at a time, and then measure the resulting drops in a pretrained model's downstream performance. **We find that models can largely recover from syntactic-style shifts, but cannot recover from vocabulary misalignment** and embedding matrix re-initialization, even with continued pretraining on 15 million tokens. Moreover, good-quality tokenizers in the transfer language do not make vocabulary alignment easier. Our experiments provide insights into the factors of cross-lingual transfer that researchers should most focus on when designing language transfer scenarios.

## 1 Introduction

What makes it hard for neural networks to learn new languages? Large language models (LLMs) require vast datasets for pretraining, making it challenging to train LLMs from scratch for low-resource languages (Devlin et al., 2018; Liu et al., 2019; Lacoste et al., 2019; Clark et al., 2020). For such languages, an appealing approach is to *transfer* knowledge from an LLM trained for a high-resource language, especially since pretrained models can transfer knowledge across even extreme shifts (Papadimitriou and Jurafsky, 2020; Tamkin et al., 2020). A range of methods have been explored to enable such crosslingual transfer of English LLMs, using techniques such as adaptive pretraining (Reimers and Gurevych, 2020), and embedding retraining (Artetxe et al., 2020; Tran, 2020). To better understand the factors affecting

---

In Chinese, "Oolong" can refer to an unexpected change or development. [*]Equal contribution. [†]Corresponding author.

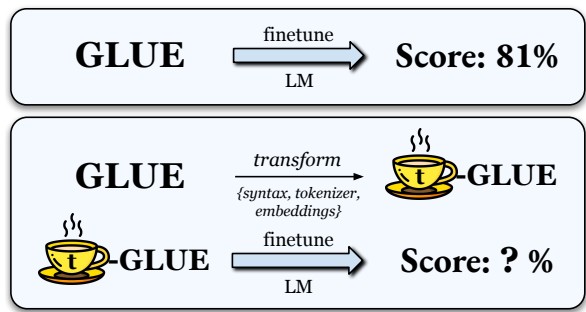

Figure 1: **Controlled transfer studies paradigm.** We systematically transform GLUE tasks (t-GLUE) to target one linguistic factor, then finetune a pretrained language model on that dataset. The resulting drop in performance indicates the importance of that factor to crosslingual transfer. See Table 1 for the list of transformations.

successful transfer, we present a set of controlled transfer studies to compare the effects of different aspects of a cross-lingual shift.

Our controlled studies consist of transferring an English model to a language that is transformed from English on just one axis of variation. Realistic transfer scenarios involve languages that differ across multiple axes of variation at one time. Our experiments serve to disentangle these effects, and identify the issues that practitioners should most focus on when doing cross-lingual transfer learning. We examine three factors that are salient in a transfer learning context:

- **Word-order syntactic differences**: Languages vary greatly in the ways that their syntax orders words. Syntactic topological similarities are generally considered an important factor when deciding transfer language pairs. We test the effects of different levels of word-order perturbation in transfer learning.

- **Word identity alignments**: Transferring to a new language requires learning the meaning, or word embeddings, of new words, and how

their layer 0 embeddings correspond to the old language. We experiment with the effect of re-initializing or shuffling the rows of the layer 0 word embedding matrix before transfer.

- **Tokenizer quality** We test the effect of bad tokenizer quality by reinitializing the word embedding matrix and transferring to English data tokenized with French and Dutch tokenizers that are suboptimal quality for English tokenization.

We test the effect of these factors on transfer learning both by 1) directly fine-tuning on t-English versions of the GLUE benchmark, as well as 2) continuing masked language model pre-training on 15 million tokens of t-English wikitext. In all cases, we find that word identity alignment provides the greatest stumbling block for transfer learning. Re-initializing or shuffling the rows of the embedding matrix has a very negative effect on downstream learning which we cannot reverse in the low-data regime that we are simulating. If the embedding matrix is reinitialized and a new tokenizer is used, the effect of reinitialization overshadows any effect that the quality of the new tokenizer might cause. In the case of syntactic word-order transformations, we find that even in the low-data transfer learning regime, the models we test can adapt to word order shifts as long as vocabulary information is kept.

We run experiments on RoBERTa, DeBERTa, and XLM-R in order to test transfer learning beyond the training set languages for both monolingual and multilingual models. Our method allows us to disentangle the effects of correlated factors by inspecting them one at a time.[1]

## 2 Related Work

As self-supervised pretraining advances the state of NLP in high-resource languages, research into widening these successes beyond high-resource languages has become widespread and important. Methodologies for best transferring a monolingual or multilingual model to an unseen language are widely explored. Ogueji et al. (2021) and Ogunremi et al. (2023), showcase the positive effects of pretraining on closer and related languages to the target language, even if this is less data than larger pretrained models, in part because of the possibility of shared vocabulary (Oladipo et al., 2022). Our

experiments build off previous efforts that try to enable crosslingual transfer from pretrained monolingual LLMs to new languages (Artetxe et al., 2018, 2020; Tran, 2020; Reimers and Gurevych, 2020; Gogoulou et al., 2021).

With respect to vocabulary sharing and adaptation, Liang et al. (2023) show that training a multilingual model with a massive vocabulary that separates out languages outweighs the benefits of vocabulary sharing between language (Patil et al., 2022), while in the transfer regime Chronopoulou et al. (2020) showcase the importance of maintaining vocabulary overlap. Techniques mapping subword embeddings to their new synonyms, or keeping subwords in the same script across languages, prove effective for cross-lingual transfer (Vernikos and Popescu-Belis, 2021; Pfeiffer et al., 2021, 2020; Muller et al., 2021). The importance of embedding intialization statistics is discussed in (Raghu et al., 2019).

Results on the importance of syntactic shifts remain broad, with work on multilingual training suggesting that syntactic shifts are significant compared to vocabulary effects (K et al., 2020), and that syntactic structure plays a role in developing parallel multilingual encodings (Dufter and Schütze, 2020), while Deshpande et al. (2022) show intersecting effects of vocabulary and word order shifts.

Understanding the direct relationship between the effect of syntactic shifts and the effect of vocabulary and tokenizer shifts remains an important problem in understanding transfer learning. Our work creates a framework for *decomposing and disentangling* the difficulties of transfer in controlled studies, giving researchers pointers for what aspects of language variation make transfer difficult.

## 3 Methods

Our methodology consists of taking a pretrained model, and transferring to a t-English: a systematically transformed version of English data that differs from English on one axis of variation. The different t-Englishes that we use are described and motivated below, and examples are in Table 1. We consider two low-data transfer environments: **Direct Fine-tuning**, where we transfer the English pretrained model directly to t-GLUE, transformed GLUE datasets (Wang et al., 2018), and **Continued Pretraining**, where we first do masked language modeling training on 15 million tokens of the WikiText-103M corpus (Merity et al., 2016)

---

[1] Our code is available publicly at `https://github.com/frankaging/oolong-crosslingual`.

| Transformation Type | Sentence / Sequence |
|---|---|
| Original English | "the film unfolds with all the mounting tension of an expert thriller , until the tragedy beneath it all gradually reveals itself ." |
| Random Order | "an all all gradually beneath thriller with reveals . until tension tragedy mounting the it of the the expert , unfolds itself film" |
| Reverse Order | ". itself reveals gradually all it beneath tragedy the until , thriller expert an of tension mounting the all with unfolds film the" |
| $\{N_{fr}, V_{fr}\}$ | "the film with all the of an expert , until the beneath all gradually . itself reveals it tragedy thriller tension mounting unfolds" |
| $\{N_{ja}, V_{ja}\}$ | "the film unfolds with all the tension of an thriller , until the tragedy beneath it all gradually itself . reveals expert mounting" |
| $\{N_{fr}, V_{ja}\}$ | "the film unfolds with all the of an expert , until the beneath all gradually . itself reveals it tragedy thriller tension mounting" |
| RoBERTa Tokenizer | "the film unfolds with all the mounting tension of an expert thriller , until the tragedy beneath it all gradually reveals itself ." |
| BERT Tokenizer | "the film un fold s with all the mounting tension of an expert thriller , until the tragedy beneath it all gradually reveals itself ." |
| Albert Tokenizer | "the film unfold s with all the mounting tension of an expert thriller , until the tragedy beneath it all gradually reveals itself ." |
| FlauBERT Tokenizer | "the film un fol ds with all the mou n ting tension of an expert thriller , un til the tr age dy bene ath it all gradu ally re ve als it self ." |
| DutchBERT Tokenizer | "the film u n f old s with all the mo unt ing te n sion of a n expert thriller , u n til the trage d y ben e ath i t all gra d u ally rev e als i t sel f ." |

Table 1: An example from the SST-2 dataset and its t-English variants. Tokenizer pre-fixes and post-fixes such as $\dot{G}$, ##, _ and $\langle /w \rangle$ are not shown for simplicity.

transformed to t-English. [2]

## 3.1 Transformed English (t-Englishes)

**Syntactic Shifts** While syntax is a crucial aspect of language (Garrett, 1976), how sensitive or invariant lanugage models are to syntactic information is a complex topic (Pham et al., 2021; Sinha et al., 2021; Papadimitriou et al., 2022; Abdou et al., 2022). In the domain of transfer learning, we investigate a set of syntactic transformations that isolate syntactic word-order shifts from the other factors that differ between languages. We bound our syntactic transformation experiments with a random shuffle control, where no word order information from the original language can be used to decode the new language. We also do the simple, but drastic baseline of reversing the order of all of the words in the input. In order to test the effect of more realistic syntactic changes, we transform the English data into t-Englishes that follow the word-order statistics of other language. Using the Galactic Dependencies package (Wang and Eisner, 2016) with Stanza (Qi et al., 2020) to transform our corpora to match the ordering of words in noun phrases and verb phrases of French ($\{\mathbf{N}_{fr}, \mathbf{V}_{fr}\}$) and Japanese ($\{\mathbf{N}_{ja}, \mathbf{V}_{ja}\}$) and also perform a mixed transformation with French noun order and Japanese verb order ($\{\mathbf{N}_{fr}, \mathbf{V}_{ja}\}$).

**Word identity alignment** Previous works have consistently found that good embeddings are crucial for enabling effective crosslingual transfer (Tran, 2020; Artetxe et al., 2020). However,

these gains may due to several factors, including better initialization statistics (Raghu et al., 2019), or to a learned alignment between the learned embeddings and the pretrained transformer layers (Wu et al., 2021). Here, we test the baseline effect of reinitializing the embedding layer while transferring to the same language that the model was pretrained. We compare this to a scenario where the rows of the embedding matrix are shuffled, meaning that vector statistics are broadly similar but each word has been swapped with another and the model needs to find the mapping during fine-tuning.

**Tokenizer** How much does tokenizer quality matter, if the price of a better tokenizer is having to reinitialize the whole word embedding matrix? Though quality tokenizers undoubtedly play an important role in multilingual NLP (Rust et al., 2020), we wish to compare the effect of tokenizer quality when the word identity alignment problem remains constant. While re-initializing the embedding matrix, we compare the effects of the original RoBERTa tokenizer, to two tokenizers that produce low-quality tokenizations for English text: the French FlauBERT (Le et al., 2020) and the Dutch DutchBERT (de Vries et al., 2019). The non-English tokenizers used to tokenize English text simulate the effect of having a bad, non-language-specific tokenizer in the low data regime (see Appendix B for statistics on how the different tokenizers work on English).

## 4 Results

We present the main results of our transfer experiments. Our experimental details (e.g. hyperparameter choices) with a per-task breakdown of t-

---

[2] For comparison, the pretraining data for RoBERTa contains 3.3B tokens, so 15M tokens is about 0.45% of its pretraining data. This is comparable to the size of the OSCAR corpus for Yiddish (Ortiz Suárez et al., 2019).

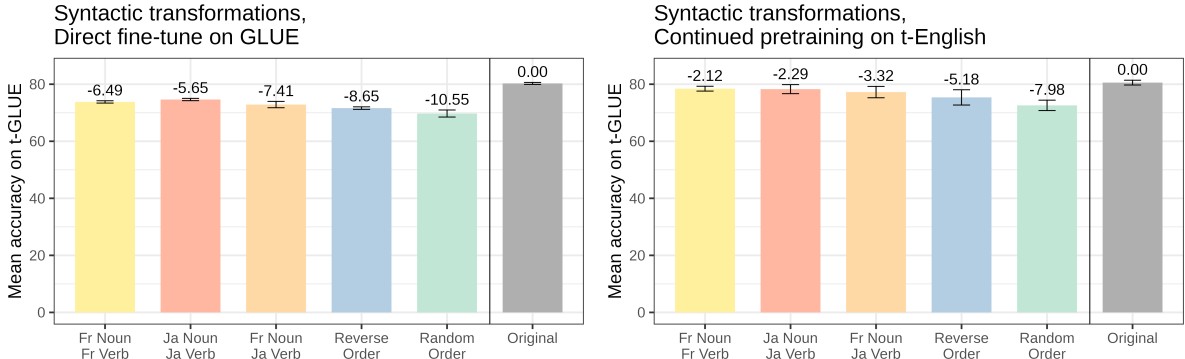

Figure 2: **Models are largely able to adapt to syntactic shifts with minor drops in performance**. Averaged GLUE scores for t-Englishes with syntactic shifts. Realistic syntactic shifts slightly impact downstream performance, while reverse and random order impact performance more significantly. Error bars represent 95% confidence intervals over 3 random seeds. Results are depicted for RoBERTa, but are consistent for all 3 models that we tested: RoBERTa, DeBERTa, and XLM-R (all results in Figure 5 in Appendix A).

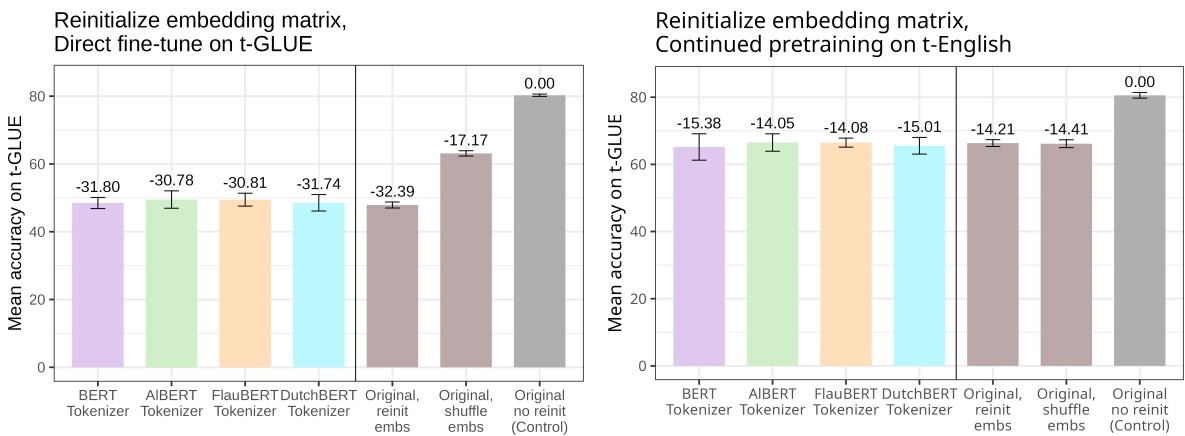

Figure 3: **Token embedding transformations are hard to recover from, regardless of tokenizer**. Averaged GLUE scores for t-Englishes with word identity perturbations. Any embedding reinitialization or shuffling, regardless of the tokenizer ultimately used, has a drastic effect on downstream performance. Error bars represent 95% confidence intervals over 3 random seeds. Results are depicted for RoBERTa, but are consistent for all 3 models that we tested: RoBERTa, DeBERTa, and XLM-R(all results in Figure 6 in Appendix A).

GLUE performance as well as additional results on DeBERTa and XLM-R are included in Appendix A.

### 4.1 Syntax matters, but training can mostly recover

Word order permutations have an effect on model performance, but the models that we test can recover relatively well from linguistic word order permutations when there are no vocabulary confounders. As shown in Figure 2, simply by fine-tuning on GLUE RoBERTa can recover from linguistic-style syntactic shifts relatively well, though this is significantly worse for random word order permutations that have no consistency or syntactic backing. These differences are all lessened

with continued pretraining on 15M tokens of the transformed t-English data. These results suggest that syntactic shifts have real but limited impact on crosslingual transfer when disentangled from vocabulary learning effects.

### 4.2 Good embeddings matter most, bad embeddings can ruin a good tokenizer

Looking at the isolated effect of vocabulary, we find that in the low-data transfer regime the model has a hard time reconstructing a reinitialized embedding matrix. As shown in Figure 3, reinitializing the embedding matrix causes huge failures for the direct fine-tune case, and the quality of the tokenizer (language-bespoke versus not) do not have an effect

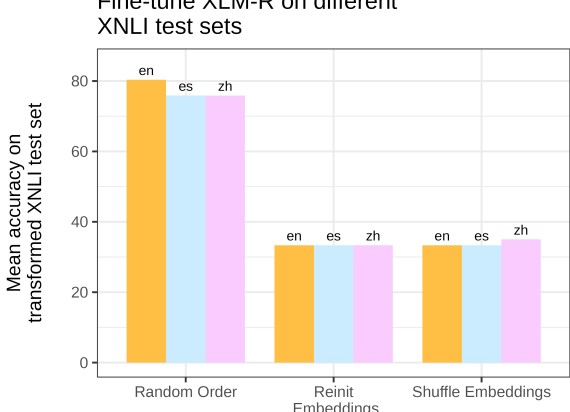

Figure 4: **Our findings generalize to fine-tuning on non-English datasets**. Fine-tuning on three different XNLI datasets yields similar findings the English GLUE findings: models can recover from the most extreme syntactic case (random ordering) much more effectively than from any of the embeddings-related perturbations. This indicates that our findings are not related to properties specific to the English language.

beyond this. Our results suggest that tokenization may thus be a "lower-order bit" for crosslingual transfer, which has little impact until good word embeddings are learned. In the direct fine-tuning case, shuffling the word embedding matrix is significantly better than reinitializing the embeddings, though this difference disappears with continued pretraining.

## 5   Conclusions

In this paper, we propose a paradigm to study crosslingual transfer through transformations which simulate and disentangle the linguistic changes across languages. Our results suggest that solving the embedding alignment problem is the "high-order bit" for crosslingual transfer: it has the largest impact on finetuning performance and is the least improved by continued pretraining. Thus, future progress on solving this problem in large-scale transformers may have outsized impact.

## Limitations

Our paper is about multilinguality in NLP. However, using multiple natural languages would make it impossible to disentangle different factors. By using controlled variants of a single language, we can create a controllable environment to investigate and understand the factors that affect real cross-lingual transfer in a multilingual setting.

Despite looking at general factors that differ between languages, and using empirical syntactic patterns from non-English languages, the fact remains that all of our experiments are centered on English and t-Englishes, and this may introduce English-centric biases.

Our scope is mainly restricted to English LLMs (vs other languages), three components of crosslingual shifts (vs other potential factors), and GLUE tasks (vs other kinds of NLP tasks). While our experiments are not an exhaustive list of linguistic properties that affect cross-lingual transfer, we aim to focus on crucial factors that change between languages, grounded by the literature. Our paradigm is extensible to other model architectures while we focus on RoBERTa in this paper with additional results on DeBERTa and XLM-R included in Appendix A.

## Ethics Statement

Our experiments provide a controlled environment to test hypotheses about what influences crosslingual transfer. However, English-based experimentations affecting other languages should not be used to determine courses of action for low-resource NLP without supplementary in-language experiments.

## Acknowledgements

We would like to thank Nelson Liu, Mirac Suzgun, and Tolúlọpẹ́ Ògúnrẹ̀mí for useful discussions and comments on drafts. This research was funded in part by NSF award number IIS-2128145.

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

# A    Results on other models

We present the results in Figures 2 and 3 for two more models: DeBERTa and the cross-lingual model XLM-R:

# B    Sequence Length Distribution

As described in Section 3.1, we try four different tokenizers to substitute for our RoBERTa (Liu et al., 2019) model that uses the Byte-Pair Encoding (BPE) (Sennrich et al., 2015) tokenizer. Specifically, we substitue with the WordPiece tokenizer (Wu et al., 2016) used by BERT (Devlin

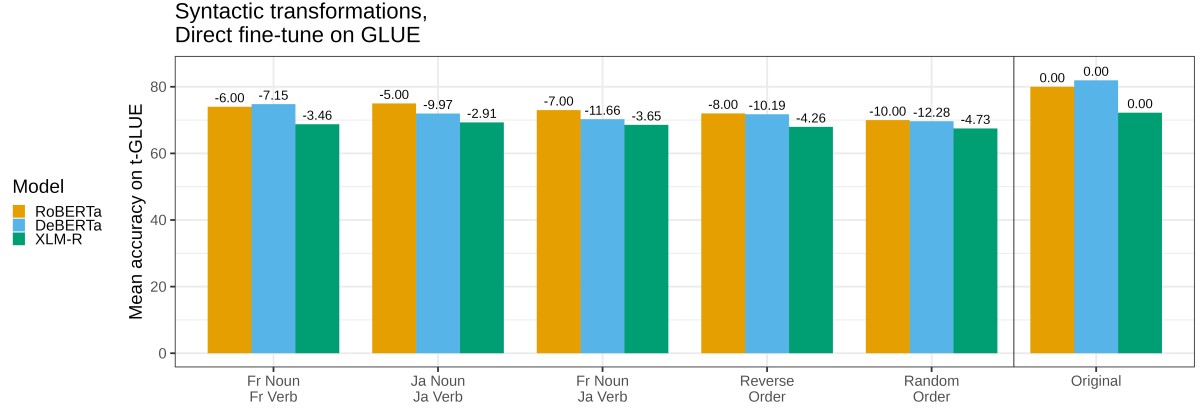

Figure 5: **Models are largely able to adapt to syntactic shifts with minor drops in performance**. Results for the embedding transformations shown for RoBERTa in Figure 2, for all models that we tested: RoBERTa, DeBERTa, and XLM-R.

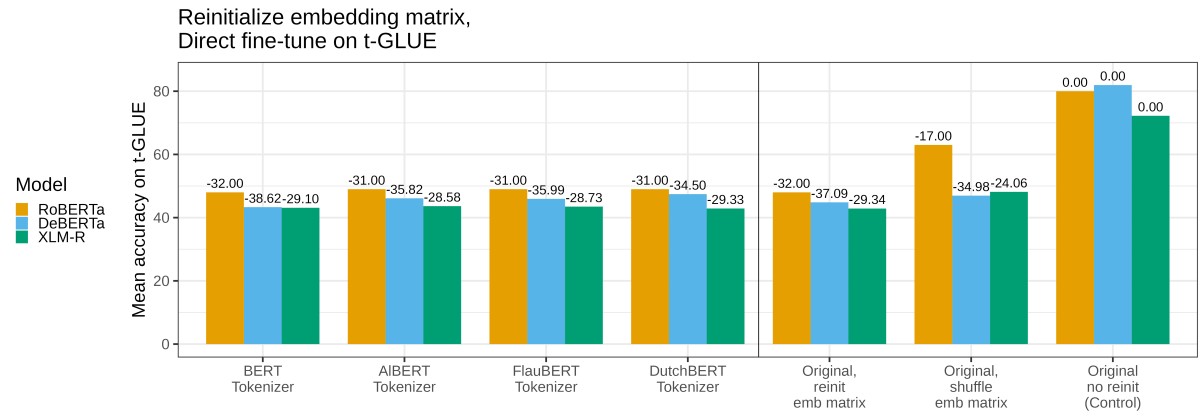

Figure 6: **Token embedding transformations are hard to recover from**. Results for the embedding transformations shown for RoBERTa in Figure 3, for all models that we tested: RoBERTa, DeBERTa, and XLM-R.

et al., 2018) (i.e., BERT Tokenizer in Table 1) and the SentencePiece tokenizer (Kudo and Richardson, 2018) used by Albert (Lan et al., 2019) (i.e., Albert Tokenizer in Table 1). Additionally, we substitute with two new non-English tokenizers including the French FlauBERT (Le et al., 2020) (FlauBERT Tokenizer in Table 1) and the Dutch DutchBERT (de Vries et al., 2019) (DutchBERT Tokenizer in Table 1). As shown in Figure 7, we plot the distributions of sequence lengths as a measure of the heterogeneity introduced by new tokenizers to ensure variences across tokenized sequence lengths. Specifically, we see there are inferior tokenizers such as FlauBERT Tokenizer with a 22.15% increase in sequence length. Our results are consistent with previous findings (Rust et al., 2020) where sequence length distributions are closer.

## C  Training Set-up Details

**Downstream Task.**  We use the GLUE benchmark (Wang et al., 2018) to evaluate model performance, which covers nine different NLP tasks. We report scores on the development sets for each task by fine-tuning our pre-trained or mid-tuned models. We fine-tune for 5 epochs for the smaller datasets (WNLI and MRPC) and 3 epochs for the others. For the performance metrics, we use Matthew's Correlation for CoLA, Pearson correlation for STS-B, and accuracy for all the other datasets.

**Hyperparameter and Infrastructure.**  For each of the mid-tuning and fine-tuning experiments, we collect averaged results from 3 runs with distinct random seeds. We tune our models with two learning rates $\{2e^{-5}, 4e^{-5}\}$, and report the best results from these two learning rates. Fine-tuning with 9 GLUE tasks takes about 8 hours on 4 NVIDIA

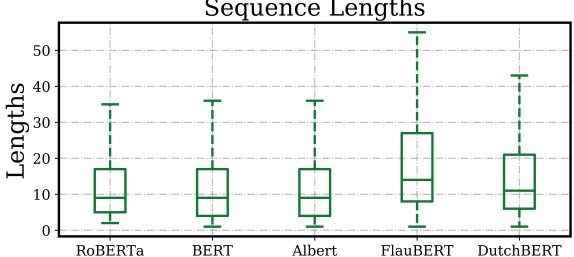

Figure 7: Distributions of sequence lengths by different tokenizers.

Titan 12G GPUs. Mid-tuning with our subset of WikiText-103M corpus takes about 18 hours with the same infrastructure.

## D    Detailed GLUE Task Performance

Table 2 shows performance break-down for individual GLUE task under different transformations as described in Section 3.1. The individual t-GLUE and GLUE results are included in Table 2. We find a consistent picture across most of the tasks, with some interesting effects like CoLA (which is more syntax-sensitive) being impacted more by syntactic shifts.

| | Original | Token Swap | Word Swap | Reinit(Emb) | Bert | Albert | FlauBERT | DutchBERT | Random | Reverse | $\{N_{fr}, V_{fr}\}$ | $\{N_{ja}, V_{ja}\}$ | $\{N_{fr}, V_{ja}\}$ |
|---|---|---|---|---|---|---|---|---|---|---|---|---|---|
| **CoLA** | .58(.01) | .00(.00) | .00(.00) | .00(.00) | .00(.00) | .00(.00) | .00(.00) | .00(.00) | .04(.05) | .01(.01) | .16(.01) | .21(.01) | .12(.01) |
| **CoLA$_{c.p.}$** | .59(.01) | .05(.07) | .02(.02) | .06(.05) | .00(.00) | .00(.00) | .01(.01) | .00(.00) | .22(.04) | .35(.01) | .45(.03) | .47(.01) | .44(.01) |
| **MNLI** | .88(.00) | .34(.01) | .50(.08) | .53(.03) | .54(.01) | .53(.01) | .67(.01) | .68(.00) | .82(.00) | .85(.00) | .86(.00) | .86(.00) | .85(.00) |
| **MNLI$_{c.p.}$** | .88(.00) | .72(.01) | .72(.01) | .73(.00) | .73(.01) | .71(.00) | .71(.01) | .69(.00) | .82(.00) | .86(.00) | .86(.00) | .86(.00) | .86(.00) |
| **MRPC** | .88(.01) | .68(.00) | .68(.00) | .68(.00) | .68(.00) | .68(.00) | .76(.01) | .77(.01) | .77(.01) | .85(.02) | .85(.01) | .86(.01) | .83(.00) |
| **MRPC$_{c.p.}$** | .87(.00) | .83(.00) | .80(.04) | .79(.01) | .82(.01) | .80(.01) | .83(.01) | .78(.00) | .81(.01) | .87(.01) | .87(.01) | .87(.01) | .86(.00) |
| **QNLI** | .93(.00) | .60(.01) | .54(.02) | .54(.04) | .55(.03) | .52(.01) | .79(.01) | .79(.00) | .88(.00) | .89(.00) | .90(.00) | .91(.00) | .90(.00) |
| **QNLI$_{c.p.}$** | .93(.00) | .83(.01) | .82(.01) | .82(.00) | .83(.00) | .82(.00) | .82(.00) | .81(.00) | .88(.00) | .91(.00) | .91(.00) | .92(.00) | .91(.00) |
| **QQP** | .91(.00) | .77(.00) | .77(.00) | .77(.00) | .76(.00) | .75(.00) | .85(.00) | .86(.00) | .90(.00) | .91(.00) | .90(.00) | .91(.00) | .90(.00) |
| **QQP$_{c.p.}$** | .91(.00) | .87(.00) | .87(.00) | .87(.00) | .87(.00) | .87(.00) | .86(.00) | .87(.00) | .90(.00) | .91(.00) | .91(.00) | .91(.00) | .91(.00) |
| **RTE** | .65(.02) | .51(.03) | .51(.03) | .53(.00) | .53(.00) | .53(.01) | .54(.02) | .56(.02) | .57(.01) | .60(.02) | .60(.01) | .61(.01) | .59(.05) |
| **RTE$_{c.p.}$** | .67(.01) | .56(.01) | .53(.01) | .54(.03) | .57(.01) | .59(.02) | .57(.03) | .57(.02) | .59(.02) | .58(.02) | .69(.01) | .64(.05) | .65(.03) |
| **SST-2** | .94(.00) | .79(.01) | .75(.02) | .79(.03) | .73(.04) | .68(.05) | .77(.01) | .78(.00) | .86(.01) | .91(.00) | .92(.00) | .92(.00) | .92(.00) |
| **SST-2$_{c.p.}$** | .94(.00) | .83(.01) | .85(.01) | .85(.01) | .83(.00) | .82(.00) | .82(.01) | .81(.01) | .88(.00) | .93(.00) | .93(.00) | .93(.00) | .92(.00) |
| **STS-B** | .89(.00) | .06(.01) | .06(.01) | .06(.02) | .09(.02) | .08(.02) | .74(.01) | .77(.00) | .87(.00) | .87(.00) | .88(.00) | .88(.00) | .88(.00) |
| **STS-B$_{c.p.}$** | .89(.00) | .76(.01) | .73(.03) | .77(.01) | .79(.01) | .78(.00) | .77(.00) | .79(.00) | .88(.00) | .87(.00) | .89(.00) | .89(.00) | .89(.00) |
| **WNLI** | .56(.00) | .56(.00) | .56(.00) | .56(.00) | .56(.00) | .58(.03) | .56(.00) | .56(.01) | .55(.01) | .56(.01) | .56(.00) | .56(.00) | .56(.01) |
| **WNLI$_{c.p.}$** | .56(.01) | .52(.06) | .53(.05) | .53(.03) | .55(.02) | .51(.07) | .56(.00) | .56(.00) | .55(.01) | .51(.07) | .56(.01) | .56(.00) | .53(.05) |

Table 2: GLUE scores for t-English with different types of interventions including scrambled word identities, syntactic shifts, and tokenizer substitutions with standard deviation (SD) for all tasks across 3 distinct runs with different random seeds. The scores with original English sentences are included for comparison. **c.p.** indicates finetuning results with continued pretrained models.