# OpenReview forum: "Oolong: Investigating What Makes Transfer Learning Hard with Controlled Studies"
_EMNLP/2023/Conference — EMNLP 2023 Main_

### Official Review · Reviewer_rSVE · 2023-08-12

**Typos Grammar Style And Presentation Improvements:** 1. L043
**Soundness:** 4

**Excitement:**

4: Strong: This paper deepens the understanding of some phenomenon or lowers the barriers to an existing research direction.

**Paper Topic And Main Contributions:**

This paper studies how different aspects of cross-lingual transfer affect transfer performance by transforming English along one aspect and investigate the results.

**Questions For The Authors:**

A. See reasons to reject

B. How was the word embedding reinitialised? Randomly? Perhaps make it clear in the text.

**Reasons To Accept:**

Clear hypothesis with sound experiments to test it

**Reasons To Reject:**

It’s unclear why only French and Japanese word orders are used to transform the English data, but this is somewhat minor

**Reproducibility:**

4: Could mostly reproduce the results, but there may be some variation because of sample variance or minor variations in their interpretation of the protocol or method.

**Reviewer Confidence:**

4: Quite sure. I tried to check the important points carefully. It's unlikely, though conceivable, that I missed something that should affect my ratings.

---

> ### Author Rebuttal · Authors · 2023-08-29
>
> We *thank* the detailed comments and suggestions. We are glad that you find our paper interesting! Here we address your detailed comments.
>
> > **Q1:** It’s unclear why only French and Japanese word orders are used to transform the English data
>
> **A1:** **The main reason is that we wanted to try a language with radically different word order to English** (Japanese is verb-final and this affects the ordering of many dependencies), as well as French which is broadly more similar to English but has some differences especially in the ordering of the modifiers of nouns. We will clarify this in our next revision.
>
> > **Q2:**  How was the word embedding reinitialized?
>
> **A2:** For each different model that we ran experiments with, **the embeddings are randomly initialized** using the provided initialization function defined with the model architecture. We will clarify this in our next revision.
>
> Lastly, we will revise our draft to correct all the typos and missing references.

---

### Official Review · Reviewer_qFy2 · 2023-08-12

**Soundness:** 3

**Excitement:**

2: Mediocre: This paper makes marginal contributions (vs non-contemporaneous work), so I would rather not see it in the conference.

**Missing References:**

- In Section 3.1, there is one missing reference: the sensitivity or invariance studies have shown surprising syntactic invariance in language models is a complex topic (Pham et al., 2021; Sinha et al., 2021; Papadimitriou et al., 2022; ?)

**Paper Topic And Main Contributions:**

This paper analyzed three factors for transferring pre-trained language models to other languages, which are word-order syntactic differences, word identity alignments, and tokenizer quality. To verify their effectiveness, the performance were measured using a transformed GLUE benchmark. To verify the word-order syntactic differences factor, the word-orders of original text were randomly shuffled, reversed, or set according to the characteristics of other languages. To verify the word identity alignment factor, the word embedding matrices were manipulated. To verify the tokenizer quality factor, the tokenizers that was not used for training were utilized. The experimental results showed that manipulating word-orders caused some performance degradation, but it can be partially recovered by fine-tuning or further pre-training. However, manipulating the word embedding matrix and replacing the tokenizer showed a large performance degradations. Thus, this paper showed that the use of well-constructed word embeddings and tokenizer selection are important factors for transferring the pre-trained language models.

**Questions For The Authors:**

- Question A: in Section 1, what are the rationale behind random word-order and reversed word-order?
- Question B: in Section 1, what are the rationale behind shuffling the rows of the layer 0 word embedding matrix?

**Reasons To Accept:**

This paper proposed three factors to be considered for transferring pre-trained language models, and verified them independently. To verify the word-order factor, the synthetic dataset was created. Creating synthetic dataset by systematically manipulating the word-order is a contribution in that it requires much less effort than constructing a dataset in a new language from scratch. To verify the word embeddings factor, a partial shuffling method was proposed in addition to a complete initialization. To verify the importance of tokenizers, different types of tokenizers that were not used in training were utilized for experiments. According to experimental results, it reveals that the cross-language differences in word-order can be resolved with further pre-training or direct fine-tuning on the target task.

**Reasons To Reject:**

This paper analyzed three factors that should be considered for the transferring pre-trained language models. The first factor, word-order, was verified with a synthetic dataset. Among the experimental settings, the word-order changes considering the characteristics of other languages is meaningful, but the reversed word-order and randomly shuffled word-order do not reflect any of cross-language characteristics and thus they will not be applicable in practice. In addition, it would be beneficial to give some real examples (not synthesized ones) where the word-order factor was applied so that readers can refer to their own cases. The second factor, word embeddings, was verified with re-initializing them and shuffling their rows. The authors need to provide sufficient justification for this. For example, what is the rationale behind shuffling rows, why not shuffle columns, etc. In addition, the shuffling rates need to be investigated to account for different level of language similarities. The third factor, tokenizer quality, was verified with various tokenizers. For cross-language transfer, French and Dutch tokenizers were utilized, but it was not verified with tokenizers of languages far from English such as Japanese.

Experimental results may only be valid for the dataset used. To show generality, authors should include experimental results for several different data sets.

**Reproducibility:**

4: Could mostly reproduce the results, but there may be some variation because of sample variance or minor variations in their interpretation of the protocol or method.

**Reviewer Confidence:**

4: Quite sure. I tried to check the important points carefully. It's unlikely, though conceivable, that I missed something that should affect my ratings.

---

> ### Author Rebuttal · Authors · 2023-08-29
>
> We *thank* the reviewer for the detailed comments and suggestions. We are glad that you find our paper interesting! Here we address your detailed comments, which have helped us further clarify the motivations behind our experiments and chart out future directions. Additionally, we will incorporate all the new results in the response into the revised version.
>
> > **Q1:**  in Section 1, what are the rationale behind random word-order and reversed word-order?
>
> **A1:**  In our word order experiments, the Japanese and French Galactic Dependency permutations attempt to capture one kind of syntactic change that would occur in a realistic transfer setting. **The random word order, on the other hand, is a lower-bound baseline which lets us see how much information is even present at all from word order.** The random word order baseline serves to put the Japanese and French numbers in context, rather than be applicable on its own. We see that recovering from a realistic syntactic transformation closes more than half the gap to this lower bound (recovering from random), which is the most difficult transformation.
>
> The random baseline is non-linguistic, but it is also not a consistent, learnable transformation. **To disentangle these two factors, we add the reverse baseline, which is still non-linguistic as you mentioned, but it is a consistent, learnable transformation.** We see that both random and reverse are worse than the naturalistic transformations, showing that the practically-relevant syntactic transformations are easier to transfer between than the worst baseline cases. Our baselines also show the surprising fact that the worst-case word-order transformation is still easier to recover from than _any_ of the vocabulary/tokenizer transformations.
>
>
> > **Q2:** in Section 1, what are the rationale behind shuffling the rows of the layer 0 word embedding matrix?
>
> **A2:** This is a great question! We tried two methods of ablating vocabulary information: reinitializing the embedding matrix and shuffling the rows (vocabulary vectors) of the embedding matrix. **We did both because we wanted to be sure that we were assessing the importance of vocabulary in the model, and not some artifact of our specific methodology.**
>
> While reinitializing the embedding is a good first step for assessing the importance of vocabulary, it puts the model in a space where all input vectors lie very close to zero and are not in the subspace that the model expects them to be in. **Compared to randomly initializing the embedding matrix, shuffling the rows lets us see whether the model is able to “reindex” the rows while the embeddings are wrong but still lie within a plausible “language subspace” for the model.** This is a more realistic test of what would happen in a crosslingual transfer scenario, when we take a model and throw out the tokenizer and fine-tune with a new tokenizer. In such a case, each row in the embedding matrix would be in a reasonable subspace and correspond to some word embedding for the old tokenizer, but would be assigned to a different word in the new tokenizer.
>
> In addition to the row-based permutation, **we also tried to add a rotation on top of the original embedding matrix, and we also found RoBERTa struggles to recover the original embeddings.** Taking these results together, LLMs are very sensitive to the “language space” laid out by the embedding matrix.
>
> > **Q3:** it was not verified with tokenizers of languages far from English such as Japanese.
>
> **A3:** This is an excellent question! **We ran additional experiments by using Chinese tokenizer from `bert-base-chinese` and evaluate crosslingual transfer across 9 tasks and with 3 different models** (RoBERTa, DeBERTa, XLM-R). Our results are shown in the following table:
>
> | Model | Original English Tokenizer | French Tokenizer | Chinese Tokenizer |
> |---------|---------|---------|---------|
> | RoBERTa  | 0.80  | 0.49  | 0.46  |
> | DeBERTa  | 0.82  | 0.46  | 0.46  |
> | XLM-R  | 0.72  | 0.43  | 0.43  |
>
> **Our results suggest that the performance drops are similar between the French tokenizer and Chinese tokenizer.** These results strengthen our conclusion that a good initialization of embeddings matters the most, rather than tokenizer quality. Note that Chinese tokenizer did result in much longer sequence length compared to other tokenizers since English words are mostly broken down to individual characters. We will discuss this set of results in detail in our next revision.
>
> > **Q4:**  Experimental results may only be valid for the dataset used.
>
> **A4:** **GLUE composes 9 tasks together**, covering a variety of domains, including sentiment analysis, NLI, and QA. This provides us with a simple yet rich enough testbed to verify our claims about crosslingual transfer. However, we agree that it would be interesting to validate this paradigm with other benchmarks, especially those non-English ones. We have started this process and reproduced some of our results.
>
> Specifically, **we ran additional experiments with 2 non-English languages** (**Spanish** and **Chinese**). We examine 4 conditions: (1) original (non-transformed); (2) reinitialized embedding matrix; (3) token identity swap (i.e., swapping rows of the embedding matrix); (4) random word order. Our results are shown in the following table:
>
> | XLM-R | Original | Reinit Embeddings | Token Swap | Random Order |
> |---------|---------|---------|---------|---------|
> | XNLI-en (English)  | 0.84  | 0.33  | 0.33  | 0.80  |
> | XNLI-es (Spanish)  | 0.81  | 0.33  | 0.33  | 0.76  |
> | XNLI-zh (Chinese)  | 0.78  | 0.33  | 0.35  | 0.76  |
>
> **Consistent with our current findings, word embeddings play a major role in crosslingual transfer for both Spanish and Chinese.** Additionally, drastic syntactic shifts only cause minimum drops in terms of performance which is what we found with our current experiments. We will provide new results on this in the next revision.
>
> Lastly, we will revise our draft to correct all the typos and missing references. Thanks a lot for your suggestions!

---

### Official Review · Reviewer_h6Fy · 2023-08-14

**Soundness:** 4

**Excitement:**

3: Ambivalent: It has merits (e.g., it reports state-of-the-art results, the idea is nice), but there are key weaknesses (e.g., it describes incremental work), and it can significantly benefit from another round of revision. However, I won't object to accepting it if my co-reviewers champion it.

**Paper Topic And Main Contributions:**

This paper experiments with systematic transformations to language in datasets to determine the importance of various language characteristics in Finetuning. This idea is particularly useful to low resource languages.

**Questions For The Authors:**

A: The work tries to assign importance to language characteristics by studying the performance of models on transformations of language. However, these "characteristics" are not impacted in the same way by Fine-tuning or even Pre-training. Can the authors explain how this generalization follows given these differences?
B: As a practical follow up to (A), 0-layer embeddings are notoriously hard to influence. To treat them on par with word-order for example, the rest of the network could be frozen (i.e, stop gradients). Would this fix the losses seen when re-initing embeddings?
C: How are sentinel tokens handled such as [CLS] when shuffling the emb-matrix? If naively shuffled it will lead to significant errors in classification / QA / GLUE tasks.
D: In random ordering of words-in-sequence, is care taken to always re-order sentences in the same way? I.e, a shuffling map is generated once and applied to all inputs? Or is input shuffling completely independent from sentence to sentence?

**Reasons To Accept:**

1. The idea is simple. Simple ideas are easier to understand and build on.
2. The work aggregates (as it should!) over multiple models to determine importance of language characteristics.
3. This is an empirical study that emulates a reasonably sized problem (even if with smaller models than SoTA).

**Reasons To Reject:**

1. Code is not public (or at least I missed the reference).  Having public code allows the research community to verify and build on these results. (EDIT: No longer true after rebuttal, as the authors have promised to release it).
2. There are several nuanced factors that are not taken into consideration (see A, B) in questions for authors.

**Reproducibility:**

4: Could mostly reproduce the results, but there may be some variation because of sample variance or minor variations in their interpretation of the protocol or method.

**Reviewer Confidence:**

4: Quite sure. I tried to check the important points carefully. It's unlikely, though conceivable, that I missed something that should affect my ratings.

**Typos Grammar Style And Presentation Improvements:**

Line 161: Million spelt wrongly
Line 170: incorrect citation ?

---

> ### Author Rebuttal · Authors · 2023-08-29
>
> We *thank* the reviewer for the detailed review! We are encouraged that you find that our method is a simple and effective way of studying crosslingual transfer. We have addressed your detailed comments below. Additionally, we will incorporate the new results in our response into the revised version.
>
> > **Q1:** Code is not public (or at least I missed the reference)
>
> **A1:** Our code can be found in the `supplementary zip file. **We will release our code for training as well as to generate t-Englishes upon publication.** Our code is largely based off HuggingFace’s training code to ensure good reproducibility of any experiment.
>
> > **Q2:** Can the authors explain how this generalization follows given these differences?
>
> **A2:** We want to clarify that t-English aims to isolate one transformation at a time (e.g., Syntactic Shifts), and study its impact on crosslingual transfer. We applied the transformation to all downstream tasks in the same way. **We agree that the impact on each task could differ, and we provided per-task breakdown in Table 2 in the Appendix.** Overall, the trend is consistent across tasks.
>
> Besides the effect on **Finetuning**, we also have parallel experiments with **Pretraining-then-Finetuning** (See bottom panel of Figure 2) where we hope the model adapts to t-Englishes with continued pretraining (i.e., standard masked language modeling). While **Pretraining-then-Finetuning** improves model performance, it does not close the gap completely, as shown through all of our experiments.
>
> > **Q3:** 0-layer embeddings are notoriously hard to influence. To treat them on par with word-order for example, the rest of the network could be frozen (i.e, stop gradients). Would this fix the losses seen when re-initing embeddings?
>
> **A3:**  This is an interesting suggestion! **We ran additional experiments by reinitializing the embedding matrix and freezing the rest of the network while training the embeddings across 9 tasks and 3 different models** (RoBERTa, DeBERTa, XLM-R). Our results are shown in the following table:
>
> | Model | Original | Reinit + Train All Layers | Reinit + Train Only the Embedding Layer |
> |---------|---------|---------|---------|
> | RoBERTa  | 0.80  | 0.48  | 0.40  |
> | DeBERTa  | 0.82  | 0.45  | 0.46  |
> | XLM-R  | 0.72  | 0.43  | 0.37  |
>
> **Our results show that tuning all layers is more effective.** Note that this is directly finetuning. In addition, we also tried this with continued pretraining on RoBERTa, and found it did not help in terms of downstream task performance. We will include these additional results in the next revision, thanks for the insightful suggestion!
>
> > **Q4:** C: How are sentinel tokens handled such as `[CLS]` when shuffling the emb-matrix?
>
> **A4:**  **We ran additional experiments where we only reinitialized the `[CLS]` token embeddings for three different models**: RoBERTa, DeBERTa, and XLM-R. We then evaluated these modifications across nine tasks. The results are tabulated below:
>
> | Model | Original | Reinit All Tokens + Train All Layers | Reinit Only `[CLS]` Tokens + Train All Layers |
> |---------|---------|---------|---------|
> | RoBERTa  | 0.80  | 0.48  | 0.78  |
> | DeBERTa  | 0.82  | 0.45  | 0.82  |
> | XLM-R  | 0.72  | 0.43  | 0.70  |
>
> Interestingly, when only reinitializing the `[CLS]` token, DeBERTa does not have any performance drops compared to other two models. Furthermore, the significant performance dips observed when reinitializing all embeddings imply that the `[CLS]` token embedding is not the only determining factor in this context.
>
> > **Q5:**  In random ordering of words-in-sequence, is care taken to always re-order sentences in the same way?
>
> **A5:**  The **random ordering experiments are designed to be the most challenging setting** where no information about ordering can be used by the network; thus, we apply different random permutations each time (i.e., training and testing orderings are different). This enables us to baseline how important non-structural information is to the model, so we can contextualize our other experiments, such as reversed ordering and reordering based on Galactic Dependencies (Table 1) which apply transformations systematically
>
> Lastly, we will revise our draft to correct the typos and missing references mentioned by the reviewer, thank you very much for all of your suggestions.

---

### Official Review · Reviewer_4H2z · 2023-08-18

**Soundness:** 4

**Excitement:**

4: Strong: This paper deepens the understanding of some phenomenon or lowers the barriers to an existing research direction.

**Paper Topic And Main Contributions:**

The paper is about studying the analysis of transformations in cross lingual transfer settings. The contribution this paper makes is to propose a controlled experiment setup to understand the important factors that can have major impact on cross-lingual transfer. The problem: Practitioners are unsure on what factors to emphasize on during cross-lingual transfer of Language Models (LMs). The solution proposed: The authors simulate a cross-lingual transfer by transforming English LM to transformed English (on one axis of variation). These variations are considered as factors that can effect Language Models. By doing one axis of variation at once, the experiments are controlled and help in analyzing the results. These factors are Word-order syntactics, word identity alignments and tokenization variations. The authors discover that word identity alignments has the major impact on a model’s capability of performing well on a different language.

**Questions For The Authors:**

q1. what sort of other factors can NLP community focus on to continue the experimentation?

**Reasons To Accept:**

1. Good experiment setup for analysis by focusing on one variation at a time. The results can be extrapolated to actual language transfer helping the NLP community for further research in multilingual models.
2. Interesting insight of knowing that ”good embeddings matter the most.”

**Reasons To Reject:**

1. As the scope of this paper was limited to English language, we can’t be certain the results would scale to other languages.
2. Inability to use the same setup for analyzing transfer of LM from one language to another. This sort of transfer experiment can further help in scaling our results to other languages.

**Reproducibility:**

3: Could reproduce the results with some difficulty. The settings of parameters are underspecified or subjectively determined; the training/evaluation data are not widely available.

**Reviewer Confidence:**

4: Quite sure. I tried to check the important points carefully. It's unlikely, though conceivable, that I missed something that should affect my ratings.

**Typos Grammar Style And Presentation Improvements:**

typos:
1. Line 17: had → hard
2. Line 43: missing “to”
3. Line 170: missing reference
4. Line 194: may → may be
5. Line 197: pretraiend → pretrained
6. Line 250: missing comma
7. Line 255: dot → do

---

> ### Author Rebuttal · Authors · 2023-08-29
>
> We *thank* the detailed comments and suggestions. We are encouraged that you find that our paper is interesting! We have addressed your detailed comments below. Here we address your detailed comments, which are helping us to revise the paper and chart out future directions. Additionally, we will incorporate all the new results in the response into the revised version.
>
> > **Q1:** “what sort of other factors can NLP community focus on to continue the experimentation?”
>
> **A1:** This is an excellent question! With our experiments focus on shifts in tokenization, syntactic shifts, and vocabulary shifts, **there are many other linguistic shifts one could investigate using our method**. For example: rewriting a language to have richer or weaker morphology (perhaps delegating the rewriting task to a language model), changing certain kinds of words but not others (e.g. to isolate the impact of articles vs nouns vs verbs), or by changing words that are most likely to change between different dialects.
>
> **In terms of additional analytical experiments, future research could look at a more fine-grained version of our experiments to gain additional insights.** For example, while we have explored extensive shifts between languages where an entirely new vocabulary is required, our t-Language methodology is versatile enough to be utilized in a broader array of contexts. For example, one could consider relearning only a smaller fraction of embeddings, either selected randomly or based on word frequency, to study a scenario approximating more closely related languages.
>
> > **Q2:** “the scope of this paper was limited to English language”
>
> **A2:** This is a great suggestion. Though our experiments are meant to simulate the effect of transferring to different languages in a controlled setting, we still only use English as the source of the t-languages. However, **we do use multilingual source models**: in the current set of experiments, we have replicated our results on English pretrained model RoBERTa (English -> t-English) on XLM-R, which is a multi-lingual model trained with more than 100 languages, as well as DeBERTa (see Appendix A.4).
>
> To address the question of non-English source languages, **we ran additional experiments with 2 non-English languages** (**Spanish** and **Chinese**). We examine 4 conditions: (1) original (non-transformed); (2) reinitialized embedding matrix; (3) token identity swap (i.e., swapping rows of the embedding matrix); (4) random word order. Our results are shown in the following table:
>
> | XLM-R | Original | Reinit Embeddings | Token Swap | Random Order |
> |---------|---------|---------|---------|---------|
> | XNLI-en (English)  | 0.84  | 0.33  | 0.33  | 0.80  |
> | XNLI-es (Spanish)  | 0.81  | 0.33  | 0.33  | 0.76  |
> | XNLI-zh (Chinese)  | 0.78  | 0.33  | 0.35  | 0.76  |
>
> **Consistent to our current findings, word embeddings play a major role in crosslingual transfer for both Spanish and Chinese.** Additionally, drastic syntactic shifts only cause minimum drops in terms of performance which is what we found with our current experiments. We will include these results in the next revision.
>
> > **Q3:** “Inability to use the same setup for analyzing transfer of LM from one language to another.”
>
> **A3:** **This is definitely right**, in the real world many factors change simultaneously between two different natural languages (e.g., English to Spanish). **However, this is one of the key motivations of our work**: our controlled experiments enable us to isolate the effect of a single factor and obtain a cleaner answer about its impact on knowledge transfer. We hope for our paper to act as an analysis of what is difficult when doing crosslingual transfer, rather than a practical setup for doing crosslingual transfer.
>
> Lastly, we will revise our draft to correct all the typos and missing references. We will release our code for training as well as the code to generate t-Englishes (currently included in our supplementary material) upon publication.

---

### Meta-Review · Area_Chair_b6DW · 2023-09-30

**Recommendation:** 4

**Metareview:**

The paper analyzes important factors in crosslingual transfer of pre-trained LLMs. The paper studies three axes of variation using controlled studies. The reviewers agree about the neat experimental set up. Reviewers raised concerns about only English language experiments but this is addressed in the rebuttal.

---

### Decision · Program_Chairs · 2023-10-07

**Decision:**

Accept-Main

**Comment:**

The paper analyzes important factors in crosslingual transfer of pre-trained LLMs. The paper studies three axes of variation using controlled studies. The reviewers agree about the neat experimental set up. Reviewers raised concerns about only English language experiments but this is addressed in the rebuttal.